# Prevalence of scabies and associated factors among children aged 5–14 years in Meta Robi District, Ethiopia

**Gemechu Ararsa[1], Emiru Merdassa[2], Tesfaye Shibiru[3], Werku Etafa[4]\***

**1** Gemechu Ararsa, West Shoa Zone Health Bureau, Oromia Region, Ambo, Ethiopia, **2** Department of Public Health, Institute of Health Sciences, Wallaga University, Nekemte, Ethiopia, **3** Department of Pediatrics and Child Health, School of Medicine, Wallaga University, Nekemte, Ethiopia, **4** Department of Pediatrics and Child Health Nursing, Institute of Health Sciences, Wallaga University, Nekemte, Ethiopia

\* witafay@gmail.com

**Data Availability Statement:** All relevant data are within the manuscript and its Supporting Information files.

**Funding:** The authors received no specific funding for this work.

## Abstract

### Background

Scabies is a public health problem that affects children and elders predominantly. Its burden is higher in resource-poor settings, and scabies has a significant impact on the long-term health of children. In Ethiopia, there is limited information about scabies in children. Therefore, the purpose of this study was to determine the prevalence of scabies and its associated factors among children aged 5–14 years in Meta Robi District, Ethiopia.

### Methods

A community-based cross-sectional study design using a multistage sampling technique was used to collect data from 457 participants by systematic random sampling. Scabies was diagnosed based on the clinical criteria set by the International Alliance for the Control of Scabies (IACS, 2020).

### Findings

The prevalence of scabies among children aged 5–14 years old was 19.26% (95%CI: 17.20–22.52). In addition, over half (54.6%) of identified cases of scabies were of moderate severity. Factors like families' low income (aOR = 2.72; 95%CI: 1.32–5.59), being a male child (aOR = 1.96, 95%CI: 1.61–4.01), using only water for hand washing (aOR = 2.01, 95% CI: 1.84–4.79), having a contact history of scabies/skin lesions (aOR = 4.15, 95%CI: 2.02–13.67), and sharing sleeping beds (aOR = 6.33, 95%CI: 2.09–19.13) were significantly associated with scabies.

### Conclusion

The study highlights a high prevalence of scabies among children aged 5–14 years in the district. Provision of adequate health education for the community and children about the scabies and delivering mass drug administration to the district is suggested.

**Competing interests:** The authors have declared that no competing interests exist.

## Introduction

Human scabies is an ectoparasitic infestation caused by the mite *Sarcoptes scabiei variety hominis* which is an obligate parasite that completes its entire life cycle on humans. Female mites burrow into the skin and lay eggs, eventually triggering a host immune response that leads to intense itching and rash [1, 2]. Scabies was listed by the World Health Organization (WHO) as a Neglected Tropical Diseases (NTDs) under category A in 2017 because it fulfills four criteria: 1) it disproportionately affects populations living in poverty; and causes important morbidity and mortality including stigma and discrimination; 2) it primarily affects populations living in tropical and sub-tropical areas; 3) it is amenable to broad control, elimination or eradication by applying public health strategies; and 4) it is relatively neglected by research. Category A NTDs require large scale action in the portfolio of WHO'S NTD Department in order to achieve control, elimination or eradication [3].

Scabies infestation occurs in all countries, but with a high burden in developing countries and tropical areas, and among infants, children, adolescents, and older persons [4, 5]. However, scabies prevalence and incidence is substantially higher in children than in adolescents and adults, and it increases sharply from five years to twenty-five years [6]. In 2015, the prevalence of scabies was from 0·2% to 71·4% in the world [4]. Findings from the Global Burden of Disease (GBD) study in 2017 reported the global prevalence and incidence cases of scabies were 175.4 million and 527.5 million, respectively, and showed that its burden is on a downward trend over the last 27 years (1990–2017) [6]. The pooled prevalence of scabies infestation was 14.5% in Ethiopia among surveyed population and it ranged from 5.5% to 23.8% among children [7]. Scabies was the most commonly reported ectoparasite in the Southwest of Ethiopia [8].

The main feature of scabies is generalized itching that is more intense during nighttime, which may lead to absenteeism from school and work, sleep disturbance that affects the quality of life, and causes stigma [9, 10]. Impetigo is skin infection caused by *Staphylococcus aureus* and *Streptococcus pyogenes*, common after persistent itching due to scabies and could complicate to severe skin and soft tissue infections, sepsis, glomerulonephritis, and acute rheumatic fever [9–11]. Immune-mediated disease and morbidity are also consequences of scabies [12]. The main route of transmission for scabies is direct skin-to-skin contact. However, crusted scabies is transmitted mainly through shared clothing or other indirect methods [13].

Scabies risk factors such as overcrowding, poor personal hygiene, sharing of beds or clothing, younger age, sex, educational status of caregivers, place of residence, poor access to water, larger family size, knowledge deficit with respect to scabies, parental illiteracy, and low-income of the households were reported [14–18]. In Ethiopia, disasters such as flooding, drought, civil war and conflict, poor water supply and sanitation, and overcrowding living conditions were the drivers of scabies [19]. Also, available scabies-related studies in Ethiopia focused on outbreak investigation [20–23]and used unmatched cases control study design [24].

Assessing the prevalence of scabies across different regions and districts of Ethiopia is important to understand the unique risk factors, to develop acceptable interventions, and beneficial to decide where resource allocation is needed. In the Western part of Ethiopia, there is no documented study on scabies and their associated factors among children aged 5–14 years in Meta Robi District, Ethiopia. Therefore, the purpose of the current study is to assess the prevalence and associated factors of scabies among children in this setting. A community-based approach for this study was utilized to include children who might not go to school.

## Methods

### Study area and period

We conducted a study in Meta Robi District, West Shoa Zone, Oromia Regional State, Ethiopia, from February 15 to March 12, 2021. Meta Robi district is around 100 kilometers from the Northwest of Addis Ababa, the capital city of Ethiopia. It is 153 kilometers far from the zonal main town, Ambo.

### Study design

This study was a cross-sectional study of scabies prevalence in selected kebeles of Meta Robi District. Randomly selected children five to fourteen years old and their parents/caregivers in selected kebeles were examined.

### Sample size and sampling procedure

We determined the sample size using the single population proportion formula by considering the following assumptions: of a 95% confidence interval, precision level of 5% [25], an expected 23.8% proportion of scabies take from the study done in Ilu Aba Bora Zone [18], and a 10% non-response rate, 307 sample size was estimated. By considering a design effect of 1.5 [26], the final sample size was found to be 461.

A multistage sampling technique was used to select the study subjects. First, Ten kebeles were selected from twenty-six kebeles by lottery. A total of 9,801 households were present in the selected ten kebeles. Next, we chose two clusters randomly from each kebele. There were 3267 households in the chosen twenty clusters. Finally, by using a systematic random sampling technique, the study subjects were recruited. A child aged 5–14 years was selected from the family assessing scabies. If there was more than one child in the selected household, we included the youngest child.

To identify the first household, we started from the central point in each selected kebele. Study direction was identified by spinning a pencil on a clipboard. Then, every seventh ($k^{th}$ = 3267/461~7) household was selected and included in the study. The total sample size was allocated proportionally to the selected villages according to their household numbers. Accordingly, the selected kebeles and their respective samples were Gurji (85), Birb, Dima (28), Fale (55), Walensu (41), Haro, Dula (42), Shino 01 (59), to Kirbe (24) and Baka (43).

### Data collection tools and techniques

Data were collected using a structured interviewer-administered questionnaire developed from the literature, and the Ethiopian Demographic and Health Survey (EDHS) was used for data collection [27] (Additional file 1). It contains five parts which include information on socio-demographic factors, water, sanitation and hygiene, and environmental-related factors, caregivers' knowledge about scabies, and health service utilization related aspects, and scabies status of children. The data collection instruments were first prepared in the English language, then translated into the local language (Afaan Oromoo), and finally retranslated back to the English language. Heads of household or household residents above 18 years old who lived with the child for more than six months and could give information were interviewed to collect the data. We recruited four healthcare workers who had previously received training about scabies to carry out the survey. Before they performed the survey, a brief training package on the basic principles of data collection and clinical diagnosis was given for two days. A trained and certified nurse supervised the study. The clinical diagnosis of scabies was based on criteria for scabies diagnosis developed by the International Alliance for the Control of Scabies (IACS, 2020) consensus criteria (Table 1)

Table 1. International Alliance for the control of scabies consensus criteria for the diagnosis of Scabies, 2020.

| Clinical category | Description | Utilized in the study |
| --- | --- | --- |
| | | (Yes or No) |
| Confirmed scabies | | |
| A1 | Mites, eggs or faeces on light microscopy of skin samples | No |
| A2 | Mites, eggs or faeces visualized on an individual using a high-powered imaging device | No |
| A3 | Mite visualized on an individual using dermoscopy | No |
| Clinical scabies | | |
| B1 | Scabies burrows | Yes |
| B2 | Typical lesions affecting male genitalia | No |
| B3 | Typical lesions in a typical distribution and two history features | Yes |
| Suspected scabies | | |
| C1 | Typical lesions in a typical distribution and one history feature | Yes |
| C2 | Atypical lesions or atypical distribution and two history feature | Yes |

[28]. The severity of scabies was based on the number of lesions counted and defined as mild (1–10 lesions), moderate (11–49 lesions) or severe scabies ($\geq$50 lesions) [29].

In this study, scabies was diagnosed clinically without any equipment. Healthcare workers were informed to refer any suspected cases of scabies or other skin lesions to a health facility for further investigation and intervention. Scabies examination was limited to easily-exposed body areas such as foot, leg to the thigh, and hand to the upper arm, scalp, and neck. A thorough body examination was not conducted as it was not practical in the field setting. Thus, a limited examination was done on common areas (hands, feet, and lower legs) where about 90% of scabies cases are detected [30]. Children wearing shoes were instructed to remove them before the examination started. Data collectors did not examine the breasts, groin, or genitals. Examinations of these sensitive areas were excluded unless requested by participants. Based on the caregiver's permission, a private examination was done in a separate place under adequate light. Data collectors wearing disposable gloves checked for the presence of scabies lesions and recorded whether there was itching, typical or atypical lesions, or a combination [28].

A pretest was done on 5% of the total sample size one week before data collection. To avoid scabies misdiagnosis, when data collectors faced the difficulty of distinguishing scabies from other skin disorders, they were instructed to take a photo of the affected skin to discuss with the supervisor in the afternoon meeting based on the assent of caregivers.

## Data analysis and processing

Data were edited and cleaned for inconsistencies and missing values, then entered into Epi Data version 3.1 and exported to SPSS version 23 for analysis. Descriptive statistics like frequency distribution were used and presented by using a table, graph, and text. Caregivers' knowledge of scabies examination was tested by five general questions from 11 points. These general questions focused on signs and symptoms of scabies, whether scabies is communicable, its route of transmission, whether it is preventable, and prevention mechanisms. These items were tested by using correct and incorrect options. Caregivers who answered correctly knowledge testing item obtained one (1) point and the rest zero (0) [17]. Caregivers' knowledge is categorized as "adequate knowledge", if caregivers scored knowledge testing items above the mean, and otherwise, they were categorized as having "inadequate knowledge" [17].

The prevalence of scabies was determined relying on the 2020 IACS Criteria subcategories B3 (clinical scabies), C1, and C2 (suspected scabies), which utilize clinical features such as typical lesions (appearance of grouped or clustered lesions, and severity and degree of secondary skin lesions due to scratching in some body areas and rubbing and atypical distribution), atypical lesion (lesions without typical morphology, or that number less than three in anybody area) and history such as itch/pruritus, and a child who has any contact with an individual diagnosed with crusted scabies or close contact with individual diagnosed with scabies, itch or typical scabies lesions in a typical distribution [28].

We used binary logistic regression analysis to determine the presence of a statistical association between scabies and independent variables. Variables with a p-value <0.25 in bivariate analysis were considered as candidates to be entered into multivariable logistic regression [31]. However, multicollinearity was checked for each pair of variables before including variables in multiple logistic regressions. Collinearity diagnostic test by Variance Inflation Factor (VIF) <5 with tolerance >0.2 was considered to have no multicollinearity problem. The necessary assumption of the logistic regression model was checked by Hosmer and Lemeshow's goodness of fit test statistics. It fits well if p-value >0.05. A p-value of<0.05 using a 95% confidence interval in multivariable logistic regressions defined a statistically significant variable.

### Ethical considerations

The study was approved by the Ethical Review Committee of Wollega University, Institute of Health Sciences (Reference number: 004CHRT/13). Permission to conduct the study was given to us by Meta Robi District Health Bureau. A written informed consent is obtained from the caregivers before they participated in the study. We explained the study object, autonomy, and confidentiality of the questionnaire, and the right to participate and withdraw at any time to both caregivers and participating children before and during collecting data.

## Results

### Socio-demographic characteristics of caregivers

A total of 457 participants enrolled in the study with a response rate of more than 99.13%. The mean age (± SD) of the involved children in the study was 9.58 ± 2.47. More than half (53.8%) of the children were from 5-to 9 years of age. Eighty-eight (19.3%) children involved in the study were attending school. Nearly three-fourths (72.9%) of the caregivers were farmers while 67.7% of them have no formal education. The average monthly income of most participants (61.9%) was more than 650 Ethiopian birr. The result showed 37.4% of the caregivers used unprotected water sources, and 79% of their traveled ≥1hour to reach a nearby health facility (Table 2).

### Characteristics and hygiene practice of children

From the total study participants, 329 (72%) of the children use water only for washing their hands, more than three quarter (75.4%) had contact history with skin itching cases/scabies and 374 (81.8%) share common sleeping bed. 75.3% had a history of contact with a person who had skin itching cases/scabies, and 81.8% shared a common sleeping bed (Table 3).

### Caregivers' source of information and knowledge about scabies

In this study, most of the participants (88%) had heard about scabies, and their main source of information was healthcare workers (35.3%). The majority of the caregivers knew at least one symptom of scabies (94.1%) and that it is a communicable disease (84.6%). Nearly 65%

**Table 2. Socio-demographic characteristics of caregivers, Meta Robi District, Ethiopia 2020.**

| Variables | Category | Frequency | Percentage |
|---|---|---|---|
| Number of children/family | ≤ 2 | 373 | 81.6 |
| | > 2 | 84 | 18.4 |
| Age of children (year) | 5–9 | 246 | 53.8 |
| | 10–14 | 211 | 46.2 |
| Sex | Male | 237 | 51.9 |
| | Female | 220 | 48.1 |
| Yes | 88 | 19.3 | Yes |
| No | 369 | 80.9 | No |
| Marital status | Together | 392 | 85.8 |
| | Separated | 65 | 14.2 |
| Occupation | Farmer | 333 | 72.9 |
| | Merchant | 58 | 12.7 |
| | Government employee | 20 | 4.4 |
| | Daily labor | 27 | 5.9 |
| | Housewife | 19 | 4.1 |
| Family size | <5 | 359 | 78.6 |
| | ≥5 | 98 | 21.4 |
| Educational status | No formal education | 309 | 67.6 |
| | Elementary school(1–8) | 93 | 20.3 |
| | Secondary school (9–2) | 35 | 7.7 |
| | College and above | 20 | 4.4 |
| Average monthly income | ≤650 ETB | 283 | 61.9 |
| | > 650 ETB | 174 | 38.1 |
| Sources of water | Piped | 91 | 19.9 |
| | Protected well or spring | 195 | 42.7 |
| | Unprotected well or spring | 171 | 37.4 |
| Household water consumption per day | <40 liters | 303 | 66.3 |
| | 41–60 liters | 98 | 21.4 |
| | >60 liters | 56 | 12.3 |
| Number of rooms in house | One | 38 | 8.3 |
| | Two | 187 | 40.9 |
| | Three | 176 | 38.5 |
| | More than three | 56 | 12.3 |
| Travel time to water source | <30 minutes | 155 | 33.9 |
| | 31–60 minutes | 224 | 49.0 |
| | >60 minutes | 78 | 17.1 |
| Travel time to nearest health facility | < 1 hour | 96 | 21.0 |
| | ≥1hour(s) | 361 | 79.0 |

reported scabies is a preventable disease and that prevention includes avoiding the sharing of fomites and beds. The overall knowledge of the caregivers was adequate (59.3%) (Table 4).

**Prevalence of scabies and its diagnostic character by 2020 IACS criteria.** The overall prevalence of scabies among school aged children (5–14 years) was 19.26% (95%CI: 17.20–22.52). Of the total cases of scabies (88), 53(60.2%) were diagnosed with a typical lesions in a typical distribution and two history features, whereas, 26 (29.6%) cases were identified with typical lesions or atypical distribution and two history feature. Among the identified scabies cases, 31(35.2%) had mild, 48 (54.6%) had moderate, and 9 (10.2%) had severe lesions (Table 5).

**Table 3. Hygiene practice of children aged 5–14 years, Meta Robi District, Ethiopia, 2020.**

| Variables | Category | Frequency | Percentage |
|---|---|---|---|
| Hand washing practiced by child | With water only | 329 | 72.0 |
| | With water and soap/local detergents | 128 | 28.0 |
| Contact history with skin itching cases/scabies | Yes | 344 | 75.3 |
| | No | 113 | 24.7 |
| Share common sleeping bed | Yes | 374 | 81.8 |
| | No | 83 | 18.2 |
| Share clothes with friends/relatives | Yes | 65 | 14.2 |
| | No | 392 | 85.8 |
| Had history of scabies diagnosed by healthcare workers | Yes | 110 | 76.0 |
| | No | 347 | 24.0 |

## Factors associated with scabies

In multivariate logistic regression, the odds of having scabies were about twice as likely among children age 5–14 years whose families' monthly income is ≤650 ETB (aOR = 2.72; 95%CI: 1.32–5.59), male children (aOR = 1.96, 95%CI: 1.61–4.01) and children who wash their hands frequently by water alone (without soap) (aOR = 2.01, 95%CI: 1.84–4.79) and more than four and six times as likely among children who had contact history of scabies/skin lesion (aOR = 4.15, 95%CI: 2.02–13.67) and share sleeping beds (aOR = 6.33, 95%CI: 2.09–19.13), respectively (Table 6).

**Table 4. Source of information and knowledge of caregivers about scabies, Meta Robi District, Ethiopia.**

| Variables | Category | Frequency | Percentage |
|---|---|---|---|
| Ever heard about scabies | Yes | 402 | 88.0 |
| | No | 55 | 12.0 |
| Source of information | Television | 40 | 10.0 |
| | Radio | 82 | 20.4 |
| | Healthcare workers | 142 | 35.3 |
| | Neighbor/family | 138 | 34.3 |
| Know at least one sign/symptom of scabies | Yes | 396 | 94.1 |
| | No | 25 | 5.9 |
| Scabies is communicable | Yes | 356 | 84.6 |
| | No | 65 | 15.4 |
| Scabies route of transmission | Skin to skin contact | 121 | 34.0 |
| | Sharing contaminated fomites and bed | 218 | 61.2 |
| | skin to skin contact and sharing fomites, bed | 17 | 4.8 |
| Scabies is preventable | Yes | 271 | 64.4 |
| | No | 150 | 35.6 |
| Scabies prevention method | Medical treatment | 76 | 28.0 |
| | Avoid skin contact | 44 | 16.2 |
| | Avoid sharing fomite and bed | 92 | 34.0 |
| | Avoid overcrowding | 13 | 4.8 |
| | Improve hygiene and sanitation | 46 | 17.0 |
| Mean knowledge score | Adequate | 271 | 59.3 |
| | Inadequate | 186 | 40.7 |

**Table 5. Prevalence of scabies and its diagnostic character by 2020 IACS criteria among children aged 5–14 years in Meta Robi District, Ethiopia, 2020 (N = 457).**

| Variables | Frequency | Percentage |
|---|---|---|
| IACS criteria | | |
| B3 | 53 | 60.2 |
| C1 | 26 | 29.6 |
| C2 | 9 | 10.2 |
| Scabies severity | | |
| Mild | 31 | 35.2 |
| Moderate | 48 | 54.6 |
| Severe | 9 | 10.2 |

## Discussion

This epidemiologic study intended to determine the prevalence of scabies and associated factors among children aged 5–14 years in Meta Robi District, Ethiopia. The overall prevalence of scabies in this district is 19.26%. Moreover, lower monthly income, being a male child, washing hands frequently with water alone (without soap), having a contact history of scabies/skin lesion, and sharing sleeping beds significantly affects scabies in the study area.

The prevalence of scabies in this study is similar to the findings in Cameroon (17.8%) and Ethiopia, Gondar town (22.5%) [15, 23]. However, it is higher than previous study findings

**Table 6. Multivariate analysis of factors associated with scabies among children aged 5–14 years in Meta Robi District, Ethiopia, 2020.**

| Variables | Category | Scabies | | Crude OR, 95% CI | Adjusted OR, 95% CI | P-value |
|---|---|---|---|---|---|---|
| | | Yes | No | | | |
| Marital status | Separated | 21(32.3%) | 44(67.7%) | 2.32 (1.29, 4.15) | 1.53 (0.68, 3.46) | > 0.05 |
| | Together | 67(17.1%) | 325(82.9%) | 1 | 1 | |
| Monthly income | ≤ 650 ETB | 72(25.4%) | 211(74.6%) | 3.37 (1.89, 6.02) | 2.72 (1.32, 5.59) | 0.01* |
| | > 650 ETB | 16(9.2%) | 158(90.8%) | 1 | 1 | |
| Child's sex | Male | 54(22.8%) | 183(77.2%) | 1.61 (1.01, 2.60) | 1.96(1. 61–4.01) | 0.004* |
| | Female | 34(15.5%) | 186(84.5%) | 1 | 1 | |
| Water source | Unprotected | 46(26.9%) | 125(73.1%) | 2.68 (1.31, 5.47) | 0.79 (0.25, 2.52) | > 0.05 |
| | Protected | 31(15.9%) | 164(84.1%) | 1.38 (0.66, 2.88) | 0.62 (0.21, 1.82) | > 0.05 |
| | Piped | 11(12.1%) | 80(87.9%) | 1 | 1 | |
| Travel time to water source | ≥30 minutes | 72(23.8%) | 230(76.2%) | 2.72 (1.52, 4.86) | 1.16 (0.54, 2.47) | > 0.05 |
| | <30 minutes | 16(10.3%) | 139(89.7%) | 1 | 1 | |
| Contact history with skin itching cases/scabies | No | 9(8.0%) | 104(92.0%) | 1 | 1 | <0.001** |
| | Yes | 79(23.0%) | 265(77.0%) | 3.45 (1.67,7.12) | 4.15 (2.02, 13.67) | |
| Share common sleeping bed/fomites | Yes | 80(21.4%) | 294(78.6%) | 2.55 (1.18, 5.51) | 6.33 (2.09, 19.13) | <0.001** |
| | No | 8(9.6%) | 75(90.4%) | 1 | 1 | |
| Washing hand with | Water only | 80(24.3%) | 249(75.7%) | 4.82 (2.26, 10.29) | 2.01 (1.84, 4.79) | 0.002* |
| | Water and soap/other local detergents | 8(6.3%) | 120(93.8%) | 1 | 1 | |
| Travel time to reach nearby health facility | < 1 hour | 13(2.8%) | 83(18.2%) | 1 | 1 | > 0.05 |
| | ≥ 1hour | 75(16.4%) | 286(62.5%) | 1.67 (0.89, 3.17) | 0.81 (0.35, 1.83) | |

Keys

**: significant at P-value < 0.001

*: significant at P-value < 0.05; OR Odds Ratio, and CI: Confidence interval

reported in Iran (3.1%), Nigeria (13.3%), Egypt (4.4%) and Côte d'Ivoire (1%) [16, 32–34]. Similarly, two previously conducted studies in Ethiopia in Yirga Cheffe (5.5%) and Dabat districts (9.3%) [8, 17] reported a lower prevalence of scabies than the current study. In contrast, the proportion of scabies reported in the Solomon Islands (54.3%), India (39.42%), and study reported in Ethiopia, in in Ilu Aba Bora Zone (23.8%) [18, 23, 29, 35]were higher than the current study finding. Studies identified personal and environmental hygiene, economic level and family size, climatic conditions (drought), and the winter season (data collection period) as the reasons for the variations in the scabies infestation rate [1, 7, 17, 18, 20, 36]. Furthermore, the difference in data collectors' quality in the clinical diagnosis of scabies, the season of the data collection, variation in the study population, and the sample size might be another possible reason for the discrepancy between the present study and previous studies.

The Ethiopian Federal Ministry of Health (FMoH) is working cooperatively with partners to rapidly stop the transmission of scabies outbreaks at the community level in high-risk districts [37]. These include scabies case management, Water, Sanitation and Hygiene (WASH), and communication for development. The Ethiopian FMoH, in villages or districts where scabies prevalence is >15%, proposed mass drug administration (MDA) except for children less than 2years, pregnant women, and lactating mothers [37]. The current descriptive finding revealed that the prevalence of scabies (19.26%) among children aged 5–14 years requires MDA in the district. The study conducted in Ethiopia that addressed a large community within a few days of MDA revealed health extension workers and community leaders as the key to the intervention [38].

The Ethiopian government has set out its development goals in successive growth and transformation plans, which identify water and sanitation as the first list of priority areas for achieving sustainable growth and poverty reduction [39]. In addition, the Ethiopian Federal Ministry of Health (FMoH) has also put in place governance mechanisms through which the performance of the hygiene and sanitation strategies can be tracked and monitored as integral parts of the health management information system [39]. Therefore, considering scabies as a water-washed disease, the key intervention is the provision of access to sufficient safe water for personal hygiene.

This study showed more than half of scabies diagnosed were moderate in severity which is harmonious with the findings from the Solomon Islands [29]. However, studies undertaken in Ghana and Liberia among the general population and in northern Ethiopia among school-aged children reported severe scabies among high proportions of study participants [17, 40, 41]. The severity of scabies infestation is directly related to the number of mites found on the skin and the lengthy of time between initial infestation and subsequent diagnosis and treatment [20, 28]. Also, scabies prevalence and severity vary depending on the season of the data collection [42]. For example, we collected data during winter. During this season, access to water for personal hygiene is challenging due to absence of the rainfall and occurrence of drought [43]. The Ethiopian Demographic and Health Survey (EDHS), 2016 report identified the water supply access coverage for rural, urban and national is 56.5%, 97.3%, and 64.85%, respectively [27]. This shows that inadequate water access coverage in the country that could expose the community at risk of communicable diseases.

According to the current study, scabies infestation is higher in males than females. This is similar to the previous studies conducted in Cameroon and Solomon Islands [15, 29]. However, it contradicts the findings in Iran [16]. Socio-cultural practices could be the possible reasons for the differences. This indicates that male children spent most of their daytime at the field playing through touching each other and handling contaminated handling articles with the scabies mite.

Also, low monthly family income was among the identified factors affecting scabies in children aged 5–14 years. Scabies more commonly affects disadvantaged populations, including

poverty, household crowding, and poor access to healthcare [1, 40]. Low-incomes expose families to scabies in two dimensions. Firstly, scabies is water-washed that can be prevented and controlled via personal hygiene practices. Families with low-income, do not use soap or other detergents for washing their hands and cloth. Secondly, families of low-income could not obtain adequate food. This primarily results in scabies infestation following reduced ability of immunity. Undernutrition affects the child's ability to mount a defense against infections or infestations and exposes them to the severest form of scabies (crust scabies).

Our analysis also showed that scabies infestation is higher among children who had a history of contact with skin itching/scabies cases similar to findings in Egypt, Nigeria, and different areas in Ethiopia [17, 21, 24, 32]. Scabies is a contagious disease mainly spread by direct and prolonged skin-to-skin contact [28]. Children share clothing with their family, family members, or friends. From the researchers' experience, most of the body parts are not covered with a cloth during the day and night time, irrespective of the weather condition (dry or raining). This could be a possible reason for high risk of scabies in children. In overcrowded or poor families, creating community awareness regarding scabies case detection to seek immediate healthcare service is better to reduce the burden of the disease in both family and the community.

Our data further revealed that children who share sleeping beds/fomites are more affected by scabies than their counterparts. However, the previously reported studies in Ethiopia did not show a significant association between scabies and shared sleeping beds/fomites [16, 21, 24, 44, 45]. Since scabies is a contagious disease [28], and the mites can exist on the clothes/articles which facilitate the transmission. Once the community has the appropriate knowledge of the scabies manifestations, transmission, and its prevention mechanism, and practices them, probably the infestation might be declined.

In the present study, children who wash their hands with water and products used for removing cleanings like soap or other local products such as plants and ash were more protected from scabies infestation. Previous studies conducted in Ethiopia in Dabat and Wadila districts also reported similar results [17, 44]. Since soap has a chemical property that removes scabies mites from the body and decreases its risk of transmission, washing hands with soap destroys mites of scabies. The study conducted in Ethiopia suggested community social and resource mobilization helps create community understanding [46]. It is critical to keep on fighting against scabies and its infestation consequences. Our analysis did not show a significant association between scabies infestation and educational levels of caregivers, family size, and knowledge about scabies.

## Conclusions

The study highlights the prevalence of scabies among children aged 5–14 years in Meta Robi district. Factors like families' low income, being a male child, using only water for hand washing, having a contact history of scabies/skin lesions, and sharing sleeping beds were significantly associated with scabies. To control scabies in the district, expansion of adequate health education for the community, and mass drug administration is suggested. Future studies using microscopic equipment used to visualize scabies mites is better for generating adequate evidences.

### Limitations of the study

This study has some limitations. First, a cross-sectional study design might not represent cause and effect relationship. The burden of scabies cases depends on the season. Our data is collected when water is limited since it was not during a rainy season, the prevalence of scabies in

the study area could be high. Second, scabies' mites are confirmed by visualizing through microscopic examination of skin samples, high-magnification devices, or dermoscopy (IACS category A, 2020). However, our diagnosis relies on clinical assessment category B (IACS, 2020). The difference in data collectors' skill and knowledge in clinical examination of scabies could be another limitation of the study as skin diseases are intertwined and difficult to distinguish from each other. We counted scabies lesions on the limited body extremities. It could affect the categorization of the severity of scabies properly. However, our study findings can be generalized to all children aged 5–14 years living in Meta Robi district because it was conducted at the community level and included children who did not attend school and based on random selection.

## Supporting information

**S1 Data.**
(SAV)

**S1 File.**
(DOCX)

**S2 File.**
(DOCX)

## Acknowledgments

We thank the study participants, the Meta Robi Health Bureau district, our data collectors, and supervisors for the effort of this study.

## Author Contributions

**Conceptualization:** Gemechu Ararsa, Emiru Merdassa.

**Data curation:** Gemechu Ararsa, Werku Etafa.

**Formal analysis:** Gemechu Ararsa, Emiru Merdassa, Tesfaye Shibiru, Werku Etafa.

**Funding acquisition:** Gemechu Ararsa.

**Investigation:** Gemechu Ararsa, Emiru Merdassa, Tesfaye Shibiru, Werku Etafa.

**Methodology:** Gemechu Ararsa, Emiru Merdassa, Tesfaye Shibiru, Werku Etafa.

**Project administration:** Gemechu Ararsa, Emiru Merdassa.

**Resources:** Gemechu Ararsa.

**Software:** Gemechu Ararsa, Emiru Merdassa, Werku Etafa.

**Supervision:** Gemechu Ararsa, Emiru Merdassa, Werku Etafa.

**Validation:** Gemechu Ararsa, Emiru Merdassa, Werku Etafa.

**Visualization:** Gemechu Ararsa, Emiru Merdassa.

**Writing – original draft:** Gemechu Ararsa, Tesfaye Shibiru, Werku Etafa.

**Writing – review & editing:** Tesfaye Shibiru, Werku Etafa.

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
