## [Decision Letter · Decision Letter 0]

25 Aug 2022

PONE-D-22-15754Prevalence of scabies and associated factors among children aged 5-14 years in Meta Robi District, EthiopiaPLOS ONE

Dear Dr. Etafa,

Thank you for submitting your manuscript to PLOS ONE. After careful consideration, we feel that it has merit but does not fully meet PLOS ONE’s publication criteria as it currently stands. Therefore, we invite you to submit a revised version of the manuscript that addresses the points raised during the review process.

We look forward to receiving your revised manuscript.

Kind regards,

Gudina Egata

Academic Editor

PLOS ONE

Journal Requirements:

Reviewers' comments:

Reviewer's Responses to Questions

**Comments to the Author**

1. Is the manuscript technically sound, and do the data support the conclusions?

Reviewer #1: Partly

2. Has the statistical analysis been performed appropriately and rigorously? 

Reviewer #1: No

3. Have the authors made all data underlying the findings in their manuscript fully available?

Reviewer #1: Yes

4. Is the manuscript presented in an intelligible fashion and written in standard English?

Reviewer #1: Yes

5. Review Comments to the Author

Reviewer #1: Dears, authors I have read this article with passion. Although the article has substance to be published, I have several suggestions and questions for further improvement before publication. Kindly find the major and minor issues. To make the review easier for authors it is a good habit to give page number and line numbers while submission. I have done by my own anyways.

Major comments

Sample size calculation

Two issues here

1. The calculated sample size do not give 279. For as low as 23.8% prevalence the use of 5% precision is unacceptable. Authors would have better used 2-3% precision that may increase the sample size.

2. Authors have used design effect of 1.5 withy out reference. Yes, design effect 1.5 to 2 may be used but the calculated design effect must be reported in the paper to appreciate the difference of the initial assumption and the actual design effect.

Why did authors use a cut of point of 0.25 during bivariable analysis? Along with it as variable screening was used initially using stepwise method after that is trivial. Enter method is automatic method to be followed if variable screening is used already.

Measurement issues: It is not clear how authors have measured some constructs such as knowledge. It is also good to at least suggest operational definition.

How variables are categorized also needs justification. How is family size categorized for example?

Line 268: The 95% CI is not provided. It makes comparison difficult. I doubt 17.8 and 13.3 may lie within the confidence interval and taking this prevalence as lower than 19.26% is trivial. We cannot judge only from the numeric values that a prevalence of 19.26 is lower than 22.5 or 23.8%. At least it should be clear that we are talking a point prevalence. As it stands now, it seems that the authors have determined the exact point estimate 100% sure, which is not scientific anyways.

Minor comments

Line 24-25, what about elders? The statement may be broken to two to make it clear.

Line 30: which type of specific random sampling technique was used?

Line 39: What did the study highlighted about prevalence of scabies as prevalent or low?

Lines 40 to 41: These are too general recommendations and come out of nowhere.

Lines 86: Need to be cited. Indicate which of these studies focus on outbreak investigation and which are institution based.

Line 163: Did authors take assent from caregivers or from children themselves?

Lines 179 to 180: citation?

Line 190: citation?

Line 201: why verbal assent, why not written?

Lines 206-207: What about in the middle of data collection?

Tables: I would suggest putting all demographic tables in one table.

Line 210: Put exact response rate? What is more than 99%? Is that 100%?

Line 264 to 266: This statement is hard conclusion. You may wish to soften it.

Lines 307 to 311: needs citation

Line 317: What does this mean? How could this be explained in terms of sample size?

Lines 320 to 323: This statement has mix up. Needs revision.

Lines 324 to 325: This is recommendation not discussion.

Lines 313-314: The case in point is, does water supply related with scabies prevalence always? How strong are evidence about this issue? Does ample water supply suffice in scabies prevalence reduction?

Lines 373 to 375: But knowledge didn’t matter in this case as evidenced earlier.

Line 373: This popped up out of nowhere. We do not know whether there was gap in adequacy of training and whether training to HEW helped in reducing scabies prevalence.

Line 375: Why is that so? Authors may comment on the use of instrument and clinical assessment based on specificity and sensitivity

6. PLOS authors have the option to publish the peer review history of their article (what does this mean?). If published, this will include your full peer review and any attached files.

Reviewer #1: No

---

## [Author Response · Author response to Decision Letter 0]

5 Sep 2022

Date: 03 September 2022

To PLOS ONE Journal 

From Werku Etafa, witafay@gmail.com

 Subject: Sending revised manuscript 

Dear Editor, 

We greatly value the time and effort you scientifically monitored and evaluated our manuscript status entitled "Prevalence of scabies and associated factors among children aged 5-14 years in Meta Robi District, Ethiopia" (manuscript number: PONE-D-22-15754). All the comments were valuable and very constructive to improve our work. We have tried to address every comment carefully and made correction which I hope could the improved the quality of the manuscript. We also highly appreciated for reviewer's work, and revised our paper point-by-point as suggested. 

Best Regards,

Werku Etafa, witafay@gmail.com

Author's point-by-point response letter

Responses to Editor

Dear editor, comments are marked in BLACK and responses are marked in RED color, pages referenced are from the final revised manuscript). 

Comments and Responses 

Response: A rebuttal letter that responds to each point raised by the academic editor and reviewer is provided, and labeled as separate file “Response to Reviewers”. 

Response: A revised manuscript with track changes is included in the current submission. 

Response: A cleaned version of manuscript labeled as “Manuscript” is uploaded. 

Journal Requirements: When submitting your revision, we need you to address these additional requirements.

Response: Thank you for updating us with the link of journal requirement. File naming is modified in the current submission. In the revised version of the manuscript, we submitted according to the journal submission requirements.

Response: All relevant data are within the paper and its supporting information files. This is reported in the current submission of the manuscript. 

Responses to Reviewer 

Dear Reviewer, 

Thank you for your concrete comments vital to improve the status of our manuscript. We appreciated and accepted your comments. Here the responses are given point-by-point below. The detail is found in the cleaned version and tracked version of the manuscript. Dear Reviewer, comments are marked in black and responses are marked in red, pages referenced in this manuscript are in the revised and cleaned version of the manuscript. Thank you once again for your appreciation and wish regarding our manuscript. 

General comments from the reviewer 

Comments 

1. Is the manuscript technically sound, and do the data support the conclusions?

Response: Thank your constructive comment; we understood that our conclusion that submitted earlier looks hard. In the revised version of our manuscript, we tried to make it soft, and recommended our suggestion as one of the interventions required (open idea). 

2. Has the statistical analysis been performed appropriately and rigorously? 

Response: Alright. One of the major mistakes made in this manuscript concerned with sample size calculation. Since the proportion of skin infestation we used in this study taken from Ilu Aba Bor Zone (P=23.8%) is less than 30, we had better to use precision level 2-3%. We agree with this point (Rule of thumb). However, there is also suggested idea to use precsion level=5% when the proportion is between 10% to 90% (Reference: Naing L, Winn T, Rusli B. Practical issues in calculating the sample size for prevalence studies. Archives of orofacial Sciences. 2006;1:9-14). We are not making an argueemnt with your suggestion, since it gives a larger sample size for generating representative evidence.

The comments regarding the reviewed work in this manuscript are categorized in to major and minor comments. We appreciated the reviewer for t

The major comments are 

Sample size calculation

Two issues here

1. The calculated sample size does not give 279. For as low as 23.8% prevalence the use of 5% precision is unacceptable. Authors would have better used 2-3% precision that may increase the sample size.

2. Response: We sorry for our mistake. The calculated sample actually gives 307 including non-response rates. Then, by considering 1.5 design effects we obtained 461 samples. This is modified in the revised version of our manuscript (page 5, line 108-112). One of the major mistakes made in this manuscript concerned with sample size calculation. Since the proportion of skin infestation we used in this study taken from Ilu Aba Bor Zone (P=23.8%) is less than 30, we had better to use precision level 2-3%. We agree with this point (Rule of thumb). However, there is also suggested idea to use precsion level=5% when the proportion is between 10% to 90% (Reference: Naing L, Winn T, Rusli B. Practical issues in calculating the sample size for prevalence studies. Archives of orofacial Sciences. 2006; 1:9-14). We are not making an argument with your suggestion, since it gives a larger sample size for generating representative evidence. 

3. Authors have used design effect of 1.5 without reference. Yes, design effect 1.5 to 2 may be used but the calculated design effect must be reported in the paper to appreciate the difference of the initial assumption and the actual design effect.

Response: Thank you for comment. We provided a reference in the revised version of the manuscript (page 5, line 111).

4. Why did authors use a cut of point of 0.25 during bivariable analysis? Along with it as variable screening was used initially using stepwise method after that is trivial. Enter method is automatic method to be followed if variable screening is used already.

Response: Here it was a typing error. We used enter method for assessing the association between one bivariable and multiple regression analysis. Thus, we used a p-value<0.25 for all candidate variables. This phrase “backward elimination” is removed from the revised manuscript. 

5. Measurement issues: It is not clear how authors have measured some constructs such as knowledge. It is also good to at least suggest operational definition.

Response: Thank you. We have revised how the score was done for knowledge assessing items and knowledge is categorized into adequate and inadequate in the revised manuscript.Caregivers' knowledge is categorized as ''adequate knowledge'', if caregivers scored knowledge testing items above the mean, and otherwise, they were categorized as having ''inadequate knowledge'' (page 9, line 179-181). 

6. How variables are categorized also needs justification. How is family size categorized for example? 

Response: Very great comment. Actually, we did not use overcrowding index for classifying family size. We classified it depending on the previous study (Dagne H, Dessie A, Destaw B, Yallew WW, Gizaw Z. Prevalence and associated factors of scabies among schoolchildren in Dabat district, northwest Ethiopia, 2018. Environmental health and preventive medicine. 2019;24(1):1-8). However, since overcrowding one of the risk factor for scabies transmission, it was better to be used overcrowding index (estimated by dividing the number of usual residents in a house by the number of bedrooms in the house). I absorbed this constructive comment for future works. 

7. Line 268: The 95% CI is not provided. It makes comparison difficult. I doubt 17.8 and 13.3 may lie within the confidence interval and taking this prevalence as lower than 19.26% is trivial. We cannot judge only from the numeric values that a prevalence of 19.26 is lower than 22.5 or 23.8%. At least it should be clear that we are talking a point prevalence. As it stands now, it seems that the authors have determined the exact point estimate 100% sure, which is not scientific anyways.

Response: I totally agree. The confidence interval (CI) for our study prevalence is (95%CI: 17.20-22.52) (Page 13, 242-43). This is included in the revised manuscript. We modified discussion when comparing our finding with others. For instance, since the CI of our study includes both the findings from Cameroon and Ethiopia at Gondar town. This is re-written in the discussion section, page 15, 271-72). 

The minor comments are 

Line 24-25, what about elders? The statement may be broken to two to make it clear.

Response: The statement is modified as “Scabies is a public health problem that affects children and elders predominantly”. 

Line 30: which type of specific random sampling technique was used?

Response:

Line 39: What did the study highlighted about prevalence of scabies as prevalent or low?

Response: We considered it high. This is stated in the revised manuscript (Page 2, Line40). 

Lines 40 to 41: These are too general recommendations and come out of nowhere.

Response: Revised as “Provision of adequate health education for the community and children about the scabies and delivering mass drug administration to the district is suggested” (page 2, line 41-42)

Lines 86: Need to be cited. Indicate which of these studies focus on outbreak investigation and which institution based are.

Response: This is cited in the revised manuscript (page 4, line 86). And the sentence after and is deleted because, it was written in mistake. 

Line 163: Did authors take assent from caregivers or from children themselves?

Response: “Assent was obtained from children aged seven years old and above after a clear explanation about the study was given to them. Children below seven years old and above participated in the study after a caregiver has provided written consent. No child was participated in the study unless the caregivers have provided a written consent.” This is written in the manuscript (page 8, line 203-205). 

Lines 179 to 180: citation?

Response: It is cited in the revised manuscript (page 8, line189) 

Line 190: citation? 

Response: It is cited in the revised manuscript (page 8, line192) 

Line 201: why verbal assent, why not written?

Response: Written assent is advisable for children greater than nine years old. For children less than nine years a written assent is used. 

Lines 206-207: What about in the middle of data collection?

Response: In the middle of data collection we also informed the data collectors to stop if the child is not calm or the families are not feeling right. 

Tables: I would suggest putting all demographic tables in one table.

Response: We summarized demographic tables together (Page 9 &10, Table 1). 

Line 210: Put exact response rate? What is more than 99%? Is that 100%?

Response: Thank you. We corrected it as “99.13%, page 9, line 211). 

Line 264 to 266: This statement is hard conclusion. You may wish to soften it.

Response: Alright. We tried to soften it (page 13, line 242-247)

Lines 307 to 311: needs citation

Response: Cited (Page 16, line 306). 

Line 317: What does this mean? How could this be explained in terms of sample size?

Response: This is removed from the sentence as it is confusing. 

Lines 320 to 323: This statement has mix up. Needs revision.

Response: This statement is revised (Page 17, line 317-318). 

Lines 324 to 325: This is recommendation not discussion.

Response: 

Lines 313-314: The case in point is, does water supply related with scabies prevalence always? How strong are evidence about this issue? Does ample water supply suffice in scabies prevalence reduction?

Response: This is revised in the manuscript submitted currently. 

Lines 373 to 375: But knowledge didn’t matter in this case as evidenced earlier.

Response: We removed this idea from the document. 

Line 373: This popped up out of nowhere. We do not know whether there was gap in adequacy of training and whether training to HEW helped in reducing scabies prevalence.

Response: Right. It is not important and removed from the manuscript. 

Line 375: Why is that so? Authors may comment on the use of instrument and clinical assessment based on specificity and sensitivity.

Response: Visualizing through microscopic examination of skin samples, high-magnification devices, or dermoscopy avoids considering diagnosis of other skin infestations as scabies. This is used to avoid over or under estimation of scabies cases.

---

## [Editor Report · Decision Letter 1]

7 Oct 2022

PONE-D-22-15754R1Prevalence of scabies and associated factors among children aged 5-14 years in Meta Robi District, EthiopiaPLOS ONE

Dear Dr. Etafa,

Thank you for submitting your manuscript to PLOS ONE. After careful consideration, we feel that it has merit but does not fully meet PLOS ONE’s publication criteria as it currently stands. Therefore, we invite you to submit a revised version of the manuscript that addresses the points raised during the review process. Please submit your revised manuscript by Nov 21 2022 11:59PM. If you will need more time than this to complete your revisions, please reply to this message or contact the journal office at plosone@plos.org. Please include the following items when submitting your revised manuscript:A rebuttal letter that responds to each point raised by the academic editor and reviewer(s). You should upload this letter as a separate file labeled 'Response to Reviewers'.A marked-up copy of your manuscript that highlights changes made to the original version. You should upload this as a separate file labeled 'Revised Manuscript with Track Changes'.An unmarked version of your revised paper without tracked changes. You should upload this as a separate file labeled 'Manuscript'.If applicable, we recommend that you deposit your laboratory protocols in protocols.io to enhance the reproducibility of your results. Protocols.io assigns your protocol its own identifier (DOI) so that it can be cited independently in the future. For instructions see: https://journals.plos.org/plosone/s/submission-guidelines#loc-laboratory-protocols. Additionally, PLOS ONE offers an option for publishing peer-reviewed Lab Protocol articles, which describe protocols hosted on protocols.io. Read more information on sharing protocols at https://plos.org/protocols?utm_medium=editorial-email&utm_source=authorletters&utm_campaign=protocols.

We look forward to receiving your revised manuscript.

Kind regards,

Gudina Egata, PhD 

Academic Editor

PLOS ONE

Journal Requirements:

Additional Editor Comments:

Dear author, no response was given to comments provided by the reviewer in line #30 and responses provided in line #201 is not clear and confusing . It is required to re-respond to these inquires before considering your work for publication.

---

## [Author Response · Author response to Decision Letter 1]

25 Oct 2022

Date: 25 October 2022

To PLOS ONE Journal 

From Werku Etafa, witafay@gmail.com

 Subject: Sending revised manuscript 

Dear Editor, 

We greatly value the time and effort you scientifically monitored and evaluated our manuscript status entitled "Prevalence of scabies and associated factors among children aged 5-14 years in Meta Robi District, Ethiopia" (manuscript number: PONE-D-22-15754R1). All the comments were valuable and very constructive to improve our work. We checked all reference list are complete and correct. We found reference number 18 and 35 as the same. We removed the reference in the list number 35. We also checked for the presence of retracted articles and there is no retracted article used to the level of our skill. We have tried to address every comment carefully and made correction which I hope could the improved the quality of the manuscript. We also highly appreciated for reviewer's work, and revised our paper point-by-point as suggested. 

Best Regards,

Werku Etafa, witafay@gmail.com

Author's point-by-point response letter

Responses to Editor

Dear editor, 

We thank you for your constructive comments. Comments are marked in BLACK and responses are marked in RED color, pages referenced are from the final revised manuscript). 

Comment: Please review your reference list to ensure that it is complete and correct. If you have cited papers that have been retracted, please include the rationale for doing so in the manuscript text, or remove these references and replace them with relevant current references. 

Response: We carefully reviewed the entire reference list and it is complete and there is modification in sequencing. Reference number 18 and Reference number 35 were the same in the previously submitted manuscript. But, we avoided this duplication in the revised version of our manuscript. We checked all reference list and we could not find a retracted article. 

Comment: Any changes to the reference list should be mentioned in the rebuttal letter that

accompanies your revised manuscript. If you need to cite a retracted article, indicate the article’s retracted status in the References list and also include a citation and full reference for the retraction notice.

Response: We included changes made in the reference list in the rebuttal letter. We did not use a retracted articles in the manuscript based on the level of our checking skill. 

Responses to Reviewer 

Comment: Dear author, no response was given to comments provided by the reviewer in line #30 and responses provided in line #201 is not clear and confusing. It is required to re-respond to these inquires before considering your work for publication.

Response: We apologize for our mistakes not providing responses to questions raised in the previous revision. For the question raised concerning line #30, we used a systematic type of random sampling (page 2, line 32). 

Comment: Responses provided in line #201 is not clear and confusing.

Response: First, we communicated with the caregivers about the study and based on their willingness a written informed consent is obtained from the caregivers before they participate in the study (page 8, line 204-205)

---

## [Editor Report · Decision Letter 2]

6 Nov 2022

Prevalence of scabies and associated factors among children aged 5-14 years in Meta Robi District, Ethiopia

PONE-D-22-15754R2

Dear Dr. Etefa _ Werku ,

We’re pleased to inform you that your manuscript has been judged scientifically suitable for publication and will be formally accepted for publication once it meets all outstanding technical requirements.

Kind regards,

Gudina Egata, PhD in Public Health

Academic Editor

PLOS ONE
---

## [Editor Report · Acceptance letter]

23 Dec 2022

PONE-D-22-15754R2 

Prevalence of scabies and associated factors among children aged 5-14 years in Meta Robi District, Ethiopia 

Dear Dr. Etafa:

I'm pleased to inform you that your manuscript has been deemed suitable for publication in PLOS ONE. Congratulations! Your manuscript is now with our production department. 

Kind regards, 

on behalf of

Dr. Gudina Egata 

Academic Editor

PLOS ONE